# Management of Medial Sphenoid Wing Meningioma Involving the Cavernous Sinus: A Single-Center Series of 105 Cases

**DOI:** 10.3390/cancers14092201

**Published:** 2022-04-28

**Authors:** Waseem Masalha, Dieter Henrik Heiland, Christine Steiert, Marie T. Krüger, Daniel Schnell, Pamela Heiland, Marco Bissolo, Anca-L. Grosu, Oliver Schnell, Jürgen Beck, Jürgen Grauvogel

**Affiliations:** 1Department of Neurosurgery, Medical Centre—University of Freiburg, 79106 Freiburg, Germany; dieter.henrik.heiland@uniklinik-freiburg.de (D.H.H.); christine.steiert@uniklinik-freiburg.de (C.S.); marie.krueger@kssg.ch (M.T.K.); pamela.heiland@uniklinik-freiburg.de (P.H.); marco.bissolo@uniklinik-freiburg.de (M.B.); oliver.schnell@uniklinik-freiburg.de (O.S.); j.beck@uniklinik-freiburg.de (J.B.); juergen.grauvogel@uniklinik-freiburg.de (J.G.); 2Faculty of Medicine, University of Freiburg, 79106 Freiburg, Germany; daniel.schnell@uniklinik-freiburg.de (D.S.); anca.grosu@uniklinik-freiburg.de (A.-L.G.); 3Department of Neurosurgery, Cantonal Hospital St. Gallen, 9000 St. Gallen, Switzerland; 4Department of Radiation Oncology, Medical Centre—University of Freiburg, 79106 Freiburg, Germany; 5German Cancer Consortium (DKTK), Partner Site Freiburg, 79106 Freiburg, Germany

**Keywords:** sphenoid wing meningioma, neurosurgery, postoperative radiotherapy, progression-free survival, cavernous sinus

## Abstract

**Simple Summary:**

Medial sphenoid wing meningiomas are among the three most common intracranial meningiomas. They present a challenge to neurosurgeons, especially when they invade critical neurovascular structures and the cavernous sinus. This study was designed to evaluate prognostic features influencing recurrence and progression-free survival of medial sphenoid meningiomas invading the cavernous sinus, with a particular focus on the impact of surgery and postoperative radiotherapy. A retrospective analysis was conducted of the database of our institution. Included were 105 cases of medial sphenoid wing meningiomas with invasion of the cavernous sinus, of which 64 were treated by surgery alone and 41 were treated by surgery plus radiotherapy. Near-total resection did reduce the risk of tumor recurrence significantly compared to subtotal resection. Progression-free survival was also significantly prolonged after postoperative radiotherapy. In conclusion, we found that performing a maximal safe resection is the factor most strongly associated with a lower recurrence rate in patients with medial sphenoid meningioma infiltrating the cavernous sinus, and that postoperative stereotactic radiotherapy of the residual tumor also significantly prolongs PFS.

**Abstract:**

Objective: Medial sphenoid wing meningiomas are among the three most common intracranial meningiomas. These tumors pose a challenge to neurosurgeons in terms of surgical treatment, as they may involve critical neurovascular structures and invade the cavernous sinus. In case of the latter, a complete resection may not be achievable. The purpose of this study was to investigate prognostic features affecting recurrence and progression-free survival (PFS) of medial sphenoid wing meningiomas involving the cavernous sinus, focusing on the contribution of surgery and postoperative radiotherapy. Methods: A retrospective analysis was conducted of the database of our institution, and 105 cases of medial sphenoid wing meningioma with invasion of the cavernous sinus, which were treated between 1998 and 2019, were included. Surgical treatment only was performed in 64 cases, and surgical treatment plus postoperative radiotherapy was performed in 41 cases. Kaplan–Meier analysis was conducted to estimate median survival and PFS rates, and Cox regression analysis was applied to determine significant factors that were associated with each therapeutic modality. Results: The risk of recurrence was significantly reduced after near-total resection (NTR) (*p*-value = 0.0011) compared to subtotal resection. Progression-free survival was also significantly prolonged after postoperative radiotherapy (*p*-value = 0.0002). Conclusions: Maximal safe resection and postoperative stereotactic radiotherapy significantly reduced the recurrence rate of medial sphenoid wing meningiomas with infiltration of the cavernous sinus.

## 1. Introduction

Intracranial meningiomas arise from the cerebral meninges and typically have benign histological features. They are the most common nonmalignant intracranial tumor (36.3%) [1]. Within meningiomas, sphenoid wing meningiomas (SWMs) account for approximately 15–20% [2]. Most neurosurgeons divide these tumors according to their anatomical location and involvement of the sphenoid wing and neighboring structures into lateral, middle, and medial tumors [3].

Despite the understanding of the anatomy of the skull base and the great progress in skull base surgery and microsurgery, medial sphenoid meningiomas (MSWMs) still pose a challenge to neurosurgeons because they invariably involve major neurovascular structures and may invade the cavernous sinus (CS). Compared to different located meningioma, these tumors have a higher morbidity, mortality, and recurrence rate, with the latter being one of the highest among all intracranial meningiomas [4].

Cavernous sinus involvement is one of the major factors influencing the extent of surgical resection in MSWMs. An aggressive resection of a meningioma invading the CS is associated with high morbidity and even mortality due to interruption of blood supply to the cranial nerves and sacrifice of or damage to the internal carotid artery [2,3,5,6]. Therefore, there is still an ongoing debate between neurosurgeons regarding whether to resect the intradural portion of the tumor but not the intracavernous portion [2,7], or to attempt a maximum safe resection of the intracavernous portion [8]. 

Studies have been conducted to evaluate this dilemma, with many describing a minimal difference between subtotal resection (STR) and gross total resection (GTR) of MSWMs infiltrating the cavernous sinus [9]. 

Some have reported similar recurrence rates after STR and GTR [6,9,10,11], whereas Mathiesen et al. reported that recurrence was significantly more frequent after STR [12].

The use of fractionated stereotactic radiotherapy also plays a critical role in the management of MSWMs infiltrating the CS, as it has been described to provide similar tumor control compared to microsurgery but with a lower procedural morbidity in terms of neurovascular damage [13,14].

Sughrue et al. performed a meta-analysis of 2065 cavernous sinus meningiomas demonstrating that radiotherapy is superior to surgical resection in terms of morbidity. However, aggressive growth of the meningioma can be observed even after years of radiotherapy [11]. The use of radiation therapy as a primary treatment modality may be compelling to some. However, there are limiting factors for its application such as the size of the tumor, the extracavernous extent of the tumor, and the proximity to radiosensitive nerve tissue. Therefore, surgical decompression is still crucial in many cases [11,14,15].

The purpose of this retrospective study was to investigate the two treatment approaches—surgery alone vs. surgery plus radiotherapy—and the impact of the different treatment approaches on progression-free survival in patients with MSWMs with invasion of the CS. In addition, prognostic factors that may affect the outcome and clinical course were investigated.

## 2. Materials and Methods

We performed a retrospective screening of all patients treated for intracranial meningiomas at the Neurosurgery Department of the Medical Center—University of Freiburg between May 1998 and June 2019. Patients lacking follow-up data were excluded from further analysis. Patient demographics, neurologic and neuro-ophthalmologic findings, radiologic results, surgical details, extent of resection, histopathologic features, adjuvant therapy, and recurrence rates were recorded and are listed in Table 1. Approval for the study was obtained from the local ethics committee of the University of Freiburg, Germany. Informed agreement was obtained from all patients.

### 2.1. Data Collection

We collected the following parameters: age at time of operation, patient sex, primary/recurrent tumors, presence of edema, pre- and postoperative KPS, cavernous sinus involvement, tumor size, extent of resection, and recurrence/progression. The mean tumor size was 40 mm^3^. Therefore, this was selected as the cutoff. MRI scans were conducted in all patients preoperatively, 3 months postoperatively, and at subsequent regular 1 year intervals. Tumor progression was defined as new lesions or a growth of residual tumor on follow-up MRI. Oncologic and neurologic outcomes were assessed using the Karnofsky Performance Scale (KPS) and clinical parameters. Patients with incomplete record data were excluded (Figure 1). Medial sphenoid wing meningiomas without infiltration of the cavernous sinus were excluded (*n* = 140) (Figure 1).

### 2.2. Extent of Resection and Surgical Approach

A microsurgical frontolateral approach was used to operate on all patients enrolled in this study at our institution. Due to the high morbidity associated with gross total resection of tumors infiltrating the cavernous sinus, surgical resection of the intracavernous portions was avoided. Intraoperative neuromonitoring of cranial nerves CN III, IV, V, and VI was performed in all surgeries. The extent of resection was determined on the basis of surgical reports and the Simpson grading scale at the 3 month follow-up MR imaging [16] and classified as follows: near-total resection (NTR), Figure 2a, in the case of a solely intracavernous residual tumor, and subtotal resection (STR), Figure 2a, in the case of intra- and extracavernous residual tumors.

### 2.3. Stereotactic Radiotherapy

Forty-one patients (39%) were treated with postoperative FSRT (fractionated stereotactic radiotherapy). The head of the patient was immobilized with a custom-made thermoplastic mask or a Brainlab stereotactic mask fixation. The ExacTrac^®^-System was used for positioning of the patient and verification before every single fraction. Alternatively, cone-beam CTs were used daily. The GTV was contoured on contrast-enhanced CT and MRI scans. The GTV included the residual tumor lesion and bony infiltration. Specifically, 0–2 mm was added to the brain and 3–4 mm was added to the dura for the CTV. It was extended by 1 mm circumferent to generate the PTV. The patients were irradiated with fractionated regimens with a median total dose of 54 Gy in 1.8 Gy single fractions. Intensity-modulated radiotherapy with the step-and-shoot technique or volumetrically modulated arc-beam therapy was usually performed. The treatment was delivered using a True-Beam Novalis 6 MeV LINAC. Treatment with modern high-precision radiotherapy was performed for WHO grade II meningioma or after subtotal resection. Patients with a postoperative KPS less than 50% were not treated with postoperative radiotherapy. The recommendation about the treatment with postoperative radiotherapy was always made after interdisciplinary discussion. Finally, a comprehensive consultation took place with patients for whom postoperative radiotherapy was recommended according to our interdisciplinary brain tumor conference. The patients themselves made the final decision.

### 2.4. Statistical Analysis

The primary endpoint in this clinical study was progression-free survival (PFS), measured as the time between surgery and tumor progression detected at MRI follow-up. Both univariate and multivariate Cox regression analyses were conducted. Only tested variables in univariate analyses with a *p*-value of less than 0.05 were included in the multivariate Cox regression analyses. There was no need to adjust the alpha level, which was set at 5%, to achieve a statistical power of at least 80%. IBM SPSS Statistics version 22 and R software tool (package: Survival, ggplot2, MANOVA) were used for all statistical analyses. The diagrams were created with the R software package ggplot2. We used the Shapiro–Wilk test (*p* > 0.05) to determine normality for distribution and variances of all data. The Wilcoxon signed rank test (unpaired) was applied for numerical variables, and the chi-square test or Fisher’s exact test were applied for nominal variables to test the difference between the groups, with an alpha level of 5%.

## 3. Results

### 3.1. Patient Clinical Data

Between 1998 and March 2019, 245 patients with MSWMs underwent surgery at our clinic, of which 131 were patients with meningioma of the medial sphenoid wing with invasion of the cavernous sinus. A total of 26 patients were not included because of lack of follow-up data (Figure 1). The mean follow-up time was 12.6 ± 6.1 years. The gender ratio (male/female) was 1:3.2 (Table 1).

As a first step, the patients were grouped according to treatment (Table 1). There were 64 patients (17 males and 47 females) in the surgery group with a median age of 64.3 years (95% confidence interval 42.5–86.1) and 41 patients (8 males and 33 females) in the surgery plus postoperative radiotherapy group with a median age of 50.9 years (95% CI 35.1–80.1). Headache, dizziness, hydrocephalus, gait disturbances, neuro-ophthalmic deficits, and cranial nerve deficits were the most common symptoms at presentation. We identified 95 cases with WHO grade I meningioma and 10 cases with WHO grade II meningioma. Patients with WHO grade II meningioma were not included in the Cox regression and survival analysis due to the dramatically different clinical course compared to grade I meningioma.

Secondly, patients were divided on the basis of tumor resection. We identified 58 (61%) patients with near-total resection and 37 (39%) patients with subtotal resection (Table 1) (Figure 2a). A tumor recurrence was observed after near-total resection in nine (9.4%) patients. In contrast, 18 patients (18.9%) showed progression after subtotal resection. Both in the Kaplan–Meier analysis (*p* = 0.0011) (Figure 2b) and in the univariate (*p* ≤ 0.0018) and multivariate analysis (*p* ≤ 0.0001), the groups differed significantly (Table 2) (Figure 3).

### 3.2. Postoperative Adjuvant Radiotherapy in WHO Grade I Meningioma

Thirty-five patients (37%) were treated postoperatively with stereotactic radiotherapy of the residual tumor in the cavernous sinus (Figure 4a), of which three patients had tumor recurrence/progression (3.1%). In contrast, 60 patients (63%) underwent surgery alone, of which 24 patients (25.2%) had tumor recurrence/progression. Both the Kaplan–Meier analysis (*p* = 0.0002) (Figure 4b) and the univariate (*p* = 0.0032) and multivariate analysis (*p* ≤ 0.0001) showed significant differences for the two groups (Figure 5) (Table 2). 

### 3.3. Other Clinical Factors

No prognostic factors affecting PFS could be identified in the additional univariate and multivariate Cox regression analysis (Table 2).

### 3.4. WHO Grade II Meningioma

We identified 10 patients with atypical medial sphenoid meningioma infiltrating the cavernous sinus. Of these, NTR could be achieved in only four patients (40%), of which two patients had tumor recurrence/progression. STR could be achieved in six patients (60%), of which five patients hat tumor recurrence/progression. Six (60%) patients were treated with postoperative radiotherapy, four of whom showed progression/recurrence. A total of four (40%) patients were not treated with postoperative radiotherapy, of which three showed progression/recurrence.

An additional Cox regression analysis was performed in patients with progressive disease; as expected, patients with meningioma WHO grade II had significantly worse PFS (*p* < 0.00001) compared to patients with WHO grade I meningioma.

### 3.5. Surgical Outcome in WHO Grade I and II Meningioma

The most frequently injured cranial nerve after surgery was the oculomotor nerve (*n* = 10, 9.5%), of which four patients had an incomplete third nerve palsy. Six patients (5.7%) had a permanent oculomotor nerve palsy.

Table 3 provides a detailed overview of all cranial nerve deficits. We also performed a chi-square test to determine a possible correlation between the resection grade and cranial nerve deficits. No correlation was found (*p* = 0.71). Additional postoperative surgery-related morbidities occurred in 11 patients (10.4%), including hydrocephalus (*n* = 1), motor weakness (*n* = 4), intracranial hematoma (*n* = 2), consciousness disorder (*n* = 1), intracranial infection (*n* = 1), and cerebrospinal fluid leak (*n* = 2). The mean preoperative Karnofsky Performance Scale (KPS) was 82% ± 15% and the mean postoperative KPS was 85% ± 13%. Pre- and postoperative KPS had no effect on PFS (Table 2).

## 4. Discussion

### 4.1. Surgery for Medial Sphenoid Wing Meningioma Invading the Cavernous Sinus

Radical surgery is considered one of the gold standards of treatment for intracranial meningiomas surpassing a diameter of >3 cm, as it has a known positive impact on PFS and overall prognosis for the patients, since the necessity to relieve neurologic symptoms is imperative in many cases [17]. However, MSWMs with infiltration of the cavernous sinus carry an associated morbidity with the surgical treatment, which excludes the possibility of performing a gross total resection. Recent neurosurgical studies have investigated this issue, and rates of gross total resection have been described between 5% and 80% [3,5,18,19]. Yet, these studies included MSWMs with and without infiltration of the CS. Our study focused on MSWMs with infiltration of the CS and aimed to investigate the role of surgical resection and postoperative radiotherapy after resection, as well as to determine possible prognostic factors for tumor recurrence. Our cohort included 105 cases of MSWM, which is one of the largest series of sphenoid wing meningiomas involving the cavernous sinus treated at a single institution published to date.

On the basis of the well-known observations in the literature, radical resection of the intracavernous portion of MSWM is often not possible without causing severe morbidity, even in Hirsch grade 0–1 [9,10,11]. Others reported that meningiomas invading the cavernous sinus of Hirsch grades 0 and 1 can be removed from the lateral compartment of the cavernous sinus with acceptable morbidity rates [6].

Our institution has a protocol to resect the tumors as safely as possible, with special attention paid to the improvement of neurological symptoms, where, e.g., a decompression of the optic nerve canal is performed to maximize visual acuity. With this in mind, a resection of the intracavernous portion of the tumors was never attempted (see Section 2). In line with this, our cohort was divided into NTR and STR groups. We observed that STR of MSWM with invasion of cavernous sinus was associated with worse PFS, when compared to NTR (Figure 2b, Table 2), which was achieved despite the risks associated with ICA tumor infiltration and involvement of further neighboring neurovascular structures. Although our study is not directly comparable to other studies in the literature, which classified the rate of GTR depending on the presence of CS invasion [4,5], included techniques for resection of the intracavernous portion of MSWMs [6], or demonstrated similar recurrence rates after GTR and STR [11], our results still highlight the importance of a better degree of resection to prolong PFS; however, this should be aimed at while maintaining functionality.

### 4.2. Functional Outcome

In our clinic, we treat the patients according to the principle of maximum and safe tumor resection, followed by adjuvant therapy, if necessary, to achieve the best functional outcome. In our series, the most common postoperative cranial nerve deficits were ocular nerves followed by the trigeminal nerve (ranging between 9.5% and 8.5%). However, some of these improved in the follow-up, after 1 year (Table 3). This is in line with other published series [6,19,20,21]. Kano et al. reported that cranial nerve deficits were found more frequently in patients who underwent primary microsurgery (92%) than in patients who underwent primary stereotactic radiosurgery (84%). However, this study included patients with small tumor volumes [15]. Chaichana et al. found in a multivariate analysis that male gender, preoperative visual deterioration, and reoperation were independent factors associated with postoperative visual deterioration [22].

In our study, no independent factor was found to predict prolonged PFS (except NTR and radiotherapy) or better clinical outcome (Table 2). Others reported similar results [2,4,10].

### 4.3. Stereotactic Radiotherapy

Stereotactic radiotherapy can be used primarily or adjuvantly for cavernous sinus meningioma, which depends on the size, location, extent, and grade of the tumor, as well as the age of the patient. Its integration in current treatment strategies is based on the similar results in terms of tumor control compared to microsurgery, but with low morbidity in terms of neurovascular injury [13,23].

However, sphenoid wing meningiomas are tumors that arise from the sphenoid wing and infiltrate secondarily into the cavernous sinus. Thus, unlike cavernous sinus meningiomas, they usually become clinically visible only when they have reached a certain size. Due to their size and proximity to sensitive neurovascular structures, this therapy modality can be limited as primary therapy [11,14,15].

Nevertheless, new radiosurgery techniques, such as robotic radiosurgery, are showing promising results [24]. A recently published meta-analysis on stereotactic radiosurgery versus stereotactic radiotherapy showed that both methods are safe options for the treatment of intracranial meningioma. However, stereotactic radiotherapy is associated with a better radiological tumor control rate and a lower incidence of symptomatic deterioration after treatment [25]. Others reported a high efficacy of stereotactic radiosurgery in the treatment of meningioma and reported long-term reliable local tumor control with low morbidity [26]. However, none of the existing studies addressed the medial sphenoid meningioma with infiltration from the cavernous sinus.

In our study, Kaplan–Meier analysis showed prolonged PFS in patients who received adjuvant radiotherapy (Figure 4b), as did the additional multivariate analysis that showed significantly better PFS (Table 2). These results are in line with other studies [11,13,14,15,27]. The meta-analysis performed by Sughrue et al. (2010a) showed the superiority of radiation treatment over surgical resection in terms of preserving cranial nerve function in cavernous sinus meningioma. Nevertheless, they emphasized the importance of long-term follow-up after radiotherapy due to the potential for aggressive growth [11]. There is strong evidence for the primarily use of stereotactic radiotherapy in meningiomas invading the cavernous sinus. However, this is limited by the size of the tumor, its extracavernous extent, and the proximity to radiosensitive vital nerve tissue. In addition, cavernous sinus meningiomas and sphenoid wing meningiomas are two types of tumors that differ in their origin, behavior, and clinical manifestation. In most cases, sphenoid wing meningiomas become clinically apparent after they reach a certain size and infiltrate the cavernous sinus. Therefore, many authors recommend surgical decompression first to reduce the proximity of these lesions to the optic nerve, diencephalon, and brainstem, followed by radiotherapy to reduce morbidity [11,14,15].

Postoperative radiotherapy for WHO grade II meningioma is still controversially discussed [28].

However, there is a consensus on the usefulness of postoperative radiotherapy after STR (Simpson grade WHO III and IV) in WHO grade II meningioma [29].

### 4.4. Limitations of the Study

This was a retrospective study performed at a single institution. Other limitations of the retrospective study design, including heterogeneous treatment strategies, differences in follow-up, and interobserver variation in resection grade assessment, need to be taken into account when interpreting the results. Nonetheless, our study is one of the largest series to date, focusing on the extent of resection of medial sphenoid wing meningiomas involving the cavernous sinus and their postoperative radiotherapy, and it validates existing findings of achieving maximum safe resection to preserve functional outcome while including radiation as an adjuvant treatment when necessary.

## 5. Conclusions

Medial sphenoid wing meningiomas involving the cavernous sinus remain a surgical challenge. The most important prognostic factor in determining recurrence is maximal safe resection, i.e., NTR.

In the present study, additional postoperative radiotherapy significantly prolonged the progression-free survival. Prospective randomized trials should be performed to establish the role of radiotherapy in the treatment of patients with meningioma of the medial sphenoid wing involving the cavernous sinus, addressing the conflicting findings from existing retrospective series.

## Figures and Tables

**Figure 1 cancers-14-02201-f001:**
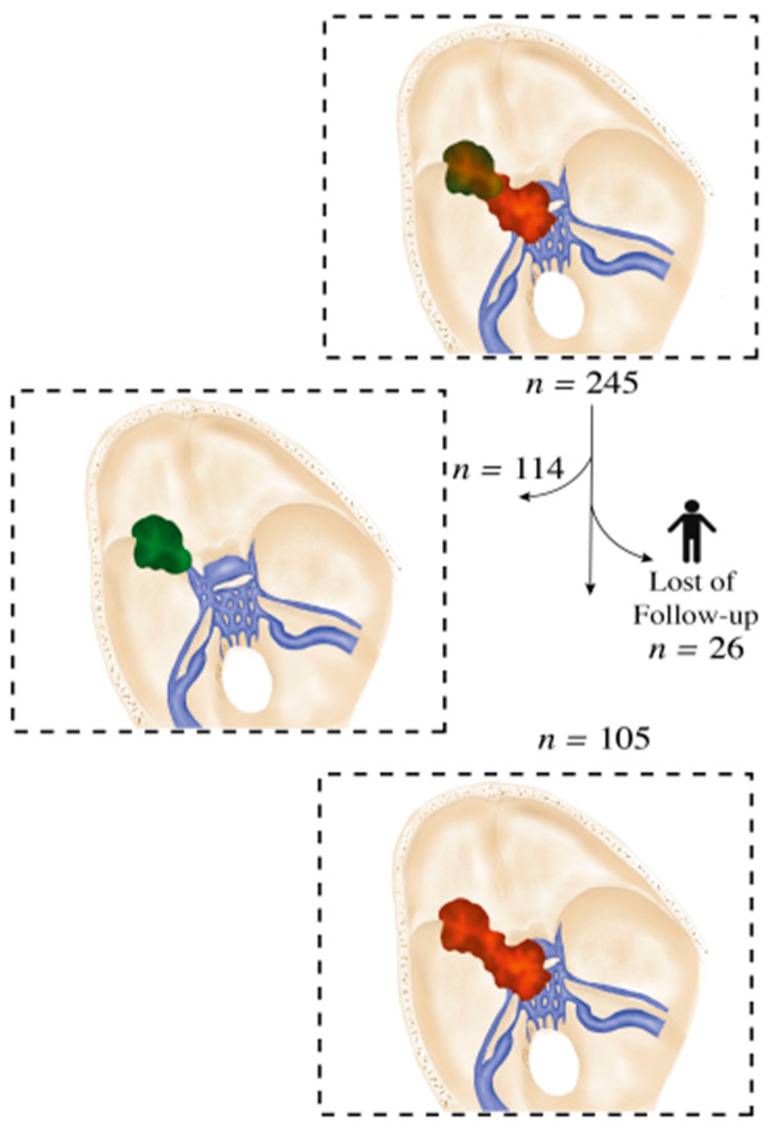
Flow diagram based on our database.

**Figure 2 cancers-14-02201-f002:**
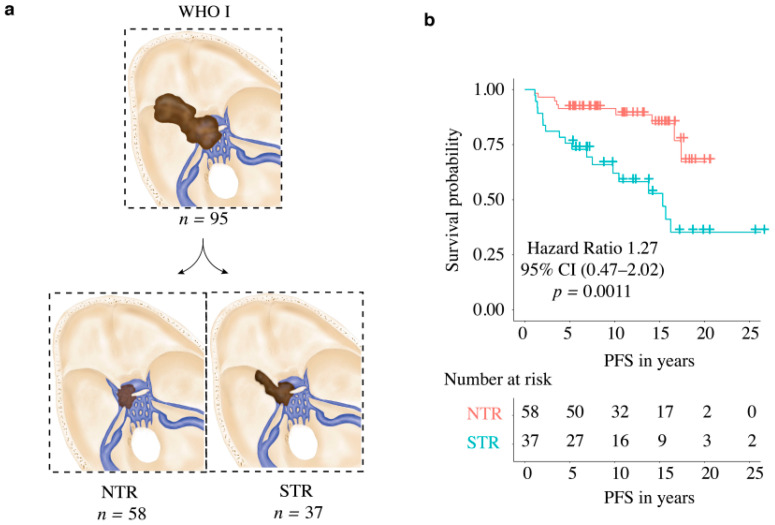
(**a**) Diagram of enrolled medial sphenoid wing meningioma WHO grade I based on resection (NTR vs. STR); (**b**) Kaplan–Meier curve: PFS in relation to extent of resection (NTR vs. STR) in WHO grade I meningioma.

**Figure 3 cancers-14-02201-f003:**
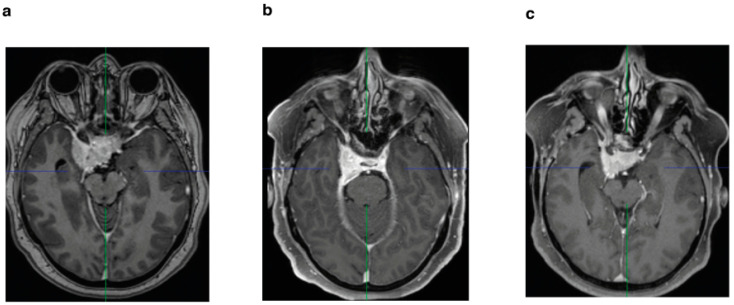
(**a**) Preoperative axial T1-weighted sequence with gadolinium enhancement of a patient who underwent STR without postoperative stereotactic radiotherapy; (**b**) postoperative MRI after 3 months; (**c**) follow-up MRI after 6 years with progression.

**Figure 4 cancers-14-02201-f004:**
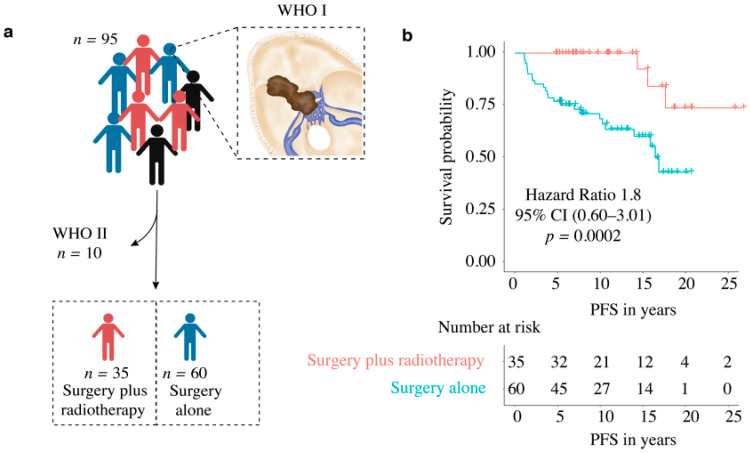
(**a**) Diagram of enrolled medial sphenoid wing meningioma based on therapy (surgery only vs. surgery plus radiotherapy); (**b**) Kaplan–Meier curve: PFS in relation to therapy (surgery only vs. surgery plus radiotherapy) in WHO grade I meningioma.

**Figure 5 cancers-14-02201-f005:**
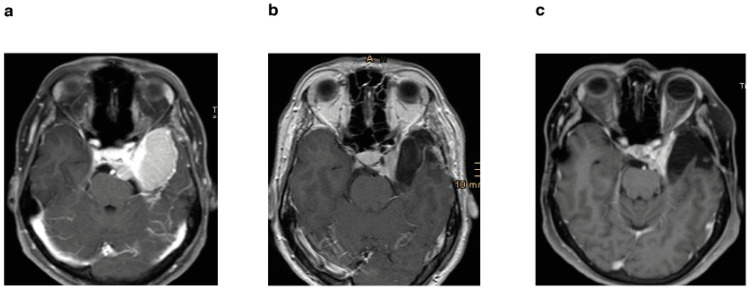
(**a**) Preoperative axial T1-weighted sequence with gadolinium enhancement of a patient who underwent STR and postoperative stereotactic radiotherapy; (**b**) postoperative MRI after 3 months; (**c**) follow-up MRI after 11 years without progression.

**Table 1 cancers-14-02201-t001:** Patient clinical data.

Parameter	Surgery *N* = 64	Surgery Plus Radiotherapy *N* = 41	*p*-Value
Age (Median, 95% CI)	64.3 (42.5–86.2)	50.9 (35.1–80.1)	*p* = 0.0024 *
Sex (*N*, %) Female Male	47 (73%) 17 (27%)	33 (80%) 8 (20%)	*p* = 0.48 **
Resection grade (*N*, %) NTR STR	44 (69%) 20 (31%)	18 (44%) 23 (56%)	*p =* 0.02 *****
Presence of edema	27 (42%)	21 (51%)	*p* = 0.42 **
Tumor Size in mm^3^ (Median, 95% CI)	32 (3–68)	22 (15–4)	*p* = 0.14 *
Primary Recurrent	53 (83%) 11 (17%)	29 (70%) 12 (30%)	*p* = 0.22 ***
WHO grade (*N*, %) WHO I WHO II	60 (94%) 4 (6%)	35 (85%) 6 (15%)	*p* = 0.39 **

* Wilcoxon test, ** Fisher’s exact test, *** chi-squared test, KPS: Karnofsky Performance Scale, CI: confidence interval, NTR: near-total resection, STR: subtotal resection.

**Table 2 cancers-14-02201-t002:** Cox regression analysis of all patients.

Variable Clinical and Treatment Factors	Progression-Free Survival		Progression-Free Survival	
	HR (95% CI)	*p*-Value	HR (95% CI)	*p*-Value
	Univariate Analysis		Multivariate Analysis	
Age (≥60 vs. <60)	0.72 (0.34–1.6)	0.41		
NTR vs. STR	3.6 (1.6–8)	0.0018	6.1 (2.7–13.7)	<0.00001
Surgery vs. surgery plus radiotherapy	6.1 (1.8–20)	0.0032	11 (3.1–36.09)	<0.00001
Primary vs. recurrent	2.2 (0.97–5.1)	0.059		
Preoperative KPS (≥80 vs. <80)	1.1 (0.5–2.5)	0.8		
Postoperative KPS (≥80 vs. <80)	1.6 (0.77–3.5)	0.2		
Presence of edema	0.71 (0.33–1.5)	0.38		
Proliferation index (≥1% vs. <1%)	1.2 (0.41–3.4)	0.76		
Tumor size (≥40 mm^3^ vs. <40 mm^3^)	0.91 (0.4–2.1)	0.82		

HR: hazard ratio, CI: confidence interval, NTR: near-total resection, STR: subtotal resection, KPS: Karnofsky Performance Scale.

**Table 3 cancers-14-02201-t003:** Clinical outcome.

Postoperative Cranial Nerve Deficits	Early (*N*, %)	Permanent (*N*, %)
I c.n	4 (3.8%)	4 (3.8%)
II c.n	4 (3.8%)	3 (2.8%)
III c.n	10 (9.5%)	6 (5.7%)
IV c.n	4 (3.8%)	2 (1.9%)
V c.n	9 (8.5%)	5 (4.7%)
VI c.n	4 (3.8%)	3 (2.8%)

c.n: cranial nerve.

## Data Availability

The data presented in this study are maintained in this article.

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
