# Peer review of "Management of Medial Sphenoid Wing Meningioma Involving the Cavernous Sinus: A Single-Center Series of 105 Cases"

_cancers, 2022, doi:10.3390/cancers14092201_

Round 1

Reviewer 1 Report

This study is to evaluate the impact of near total resection (NTR) or subtotal resection (STR), as well as the additional postoperative radiotherapy on progression-free survival in medial sphenoid wing meningioma patients with the invasion of the cavernous sinus. The rationale is reasonable and the objectives are clearly defined. The findings are well-presented and the investigation methods are described thoroughly. The only question that I have is whether there is any correlation of NTR or STR to the deficits of the cranial nerves described in Table 3. Despite a better PFS, it would be more interesting to see if NTR did not cause more damage to the patients.

Author Response

First, we want to thank you for this constructive question.

We performed a chi-square test to determine a possible correlation between the degree of resection and cranial nerve deficits. No correlation was found (p=0.71).

We added this to the results section (Line 253-255)

Reviewer 2 Report

The authors present an interesting manuscript on skull base meningiomas.

Some aspects should be considered before accepting the manuscript for publication:

-line 50 "otherwise located" please rephrase

-how is age a limiting factor against radiotherapy (and pro neurosurgery?)

-What do you mean by "secondary" meningioma (Table 1)? I guess you infer recurrent meningioma? The term "secondary" rather infers metastatic disease and I suggest rephrasing of these parameters.

-"In our institution the treatment with modern high-precision radiotherapy was performed depending on the WHO grade, postoperative KPS and the extent of resection." Please describe in more detail as this is crucial to the findings of the study: e.g. only WHO2-3 tumors? all WHO2 tumors?

-Please use one style of significance presentation e.g. P=0.XXX throughout the manuscript

-l 193: "the most affected" I guess in terms of frequency and not in degree of affection? Please clarify

-had "an" incomplete deficit "l 194"

-l 218: goal standards or gold standard or goal? Please revise

-Please omit p-values from the discussion

-line 279 stereotactic instead of Stereotactic

-Conclusion: please limit your conclusion to data provided by your results - you did not investigate radical approaches, hence this aspect should be dropped from the conclusion. "the use of radiotherapy becomes increasingly more common" how do you justify this conclusion based on your data?

-The introduction and the further direction of the manuscript do not fit well: while the introduction suggests a study of radiotherapy vs. surgery in the treatment, this study rather touches a different question surgery vs. surgery+radiotherapy. Please revise your introduction

-during the study period, how many patients underwent radiotherapy/radiosurgery without prior surgery and why? How significant would you consider this selection bias?

-You excluded patients in cases of incomplete data availability. Fig 1 presents 26 patients lost to follow-up. Are these patients included?

-Did you follow-up by phone/mail for the purpose of this study? If not, how did you define "lost to follow-up"?

-You report a substantial portion of patients without follow-up. How was the neurological status of these patients at discharge? 

Author Response

The authors present an interesting manuscript on skull base meningiomas.

Some aspects should be considered before accepting the manuscript for publication:

Thank you very much for your constructive comments.

Below we have the corrections you requested

-line 50 "otherwise located" please rephrase

As required, we have rephrased the above-mentioned term (line 95)

-how is age a limiting factor against radiotherapy (and pro neurosurgery?)

Sughrue et al. and others have shown that aggressive growth of meningiomas, which are primarily benign tumors, can be observed even after years of radiotherapy. Therefore, probably it is preferable in a young patient who still has a long life expectancy to wait after surgical treatment rather than immediately perform radiotherapy, and in case of progression, radiotherapy can be discussed again.

we have taken age out of the text as a limiting factor (line 115-117)

-What do you mean by "secondary" meningioma (Table 1)? I guess you infer recurrent meningioma? The term "secondary" rather infers metastatic disease and I suggest rephrasing of these parameters.

Thank you for this comment, we have changed the term secondary to recurrent in table 1 and 2 as requested

-"In our institution the treatment with modern high-precision radiotherapy was performed depending on the WHO grade, postoperative KPS and the extent of resection." Please describe in more detail as this is crucial to the findings of the study: e.g. only WHO2-3 tumors? all WHO2 tumors?

Thank you for this important comment

we have described the indication for postoperative radiotherapy more precisely (line182-188).

Ultimately, the indication for postoperative radiotherapy was always discussed by an interdisciplinary team.

In our series there were no patients with meningioma WHO III.

-Please use one style of significance presentation e.g. P=0.XXX throughout the manuscript

We checked the manuscript and unified the style of significance presentation

-l 193: "the most affected" I guess in terms of frequency and not in degree of affection? Please clarify

we rewrote the sentence, Line 250

-had "an" incomplete deficit "l 194"

We mean Incomplete third nerve palsy. We changed the Term ‘incomplete deficit’ to ‘Incomplete third nerve palsy’ (Line 251-252)

-l 218: goal standards or gold standard or goal? Please revise

We mean gold standard.

-Please omit p-values from the discussion

p-value is omited from the discussion

-line 279 stereotactic instead of Stereotactic

we changed the tip error

-Conclusion: please limit your conclusion to data provided by your results - you did not investigate radical approaches, hence this aspect should be dropped from the conclusion. "the use of radiotherapy becomes increasingly more common" how do you justify this conclusion based on your data?

We rewrote the conclusion section and limited our conclusion to the data providing our results, as requested (line 353-359).

-The introduction and the further direction of the manuscript do not fit well: while the introduction suggests a study of radiotherapy vs. surgery in the treatment, this study rather touches a different question surgery vs. surgery+radiotherapy. Please revise your introduction

We rewrote the aim of the study in lines 119-122

-during the study period, how many patients underwent radiotherapy/radiosurgery without prior surgery and why? How significant would you consider this selection bias?

Thank you for this very important comment

in our series there were no patients who were treated only with radiotherapy.

It is because medial sphenoid wing meningiomas that infiltrate the cavernous sinus arise primarily from the sphenoid wing and infiltrate the cavernous sinus secondarily. In contrary to the cavernous sinus meningioma, they usually become clinically apparent only when they have infiltrated the cavernous sinus and have reached a certain size. Due to their size and proximity to sensitive neurovascular structures, radiotherapy as primary therapy may be limited.

We explained this point and the difference to cavernous sinus meningioma in the discussion (line 311-316 and 333-336)

-You excluded patients in cases of incomplete data availability. Fig 1 presents 26 patients lost to follow-up. Are these patients included?

These 26 patients were not included in the analysis.

Most of these patients were foreign patients who had surgery in our clinic and then travelled back to their home countries. tracking these patients was not possible.

Others did not present themselves to our clinic for a follow-up for unknown reasons.

-Did you follow-up by phone/mail for the purpose of this study? If not, how did you define "lost to follow-up"?

The follow-up takes place in our clinic through a presentation of the patients in person. at the same time, the patients undergo an examination and a control MRI is carried out.

Generally, the patients are presented 3 months after the surgery and then annually for clinical and MRI checks.

we mentioned it in line 155-156

-You report a substantial portion of patients without follow-up. How was the neurological status of these patients at discharge? 

Below we present a table with the 26 patients with lost of follow up and their neurological status after the operations. The patients are anonymised

Patient

Postoperative cranial nerves deficits

Preoperative KPS

Postoperative KPS

1

none

70

80

2

none

90

90

3

none

unknown

70

4

3, 5

90

80

5

none

80

80

6

none

90

90

7

none

80

80

8

3

80

80

9

none

70

80

10

none

80

80

11

none

90

90

12

none

60

80

13

none

90

80

14

4,5,6

80

70

15

none

90

80

16

none

90

90

17

none

80

80

18

none

unknown

70

19

5

90

90

20

none

70

70

21

none

80

90

22

3

90

80

23

none

60

unknown

24

none

90

90

25

none

unkonwn

70

26

none

70

80

Reviewer 3 Report

This is a valuable manuscript for the treatment of medial sphenoidal wing meningiomas involving the cavernous sinus. I would like it if radiosurgery were better discussed in contrast to stereotactic radiation.

Author Response

This is a valuable manuscript for the treatment of medial sphenoidal wing meningiomas involving the cavernous sinus. I would like it if radiosurgery were better discussed in contrast to stereotactic radiation.

First of all, thank you for your positive feedback about our studies and for the constructive comment.

We added more detail about radiosurgery in the discussion (line 318-324).

Reviewer 4 Report

The authors present a single-center, retrospective analysis of sphenoid wings meningiomas invading the cavernous sinus (CS), submitted to surgery between 1998 and 2019. This is one of the largest series of such diseases presented to date, and I think that the manuscript could represent a relevant addition to the existing literature. 

As the authors could imagine, my main issue is related to the protocol based on never attempting to remove the intracavernous portion of the tumors: considering that, as also fig 2 depicts, the EOR has an impact on PFS (regardless of adjuvant radiotherapy), did the authors never try to remove the intracavernous part of meningiomas, also in case of tumor grading modified Hirsch Grades 0 and 1? 

Author Response

First of all, thank you for your positive feedback and your very important comment.

As you know as a surgeon, resection of the intracavernous portions, even in Hirsch grade 0-1, is always associated with a slightly increased risk. For us, functionality was very important, so we did not try to remove these parts.

However, we think that this aspect should be mentioned.

We have mentioned it in the discussion (277-279).

Round 2

Reviewer 2 Report

The authors have appropriately addressed my concerns and reservations.